# Differential Methylation Profile in Fragile X Syndrome-Prone Offspring Mice after in Utero Exposure to Lactobacillus Reuteri

**DOI:** 10.3390/genes13081300

**Published:** 2022-07-22

**Authors:** Reem R. AlOlaby, Marwa Zafarullah, Mariana Barboza, Gang Peng, Bernard J. Varian, Susan E. Erdman, Carlito Lebrilla, Flora Tassone

**Affiliations:** 1Department of Mathematics & Sciences, College of Health Sciences, California Northstate University, Sacramento, CA 95670, USA; reem.alolabi@cnsu.edu; 2Biochemistry and Molecular Medicine, University of California, Davis, Sacramento, CA 95817, USA; mzafarullah@ucdavis.edu (M.Z.); mbarboza@ucdavis.edu (M.B.); clebrilla@ucdavis.edu (C.L.); 3Department of Biostatistics, Yale University School of Public Health, New Haven, CT 06510, USA; gang.peng@yale.edu; 4Division of Comparative Medicine, Massachusetts Institute of Technology, Cambridge, MA 02139, USA; bvarian@mit.edu (B.J.V.); serdman@mit.edu (S.E.E.); 5MIND Institute, UC Davis, Sacramento, CA 95817, USA

**Keywords:** FXS, in utero, methylation, epigenetics, *Lactobacillus reuteri*, ASD

## Abstract

Environmental factors such as diet, gut microbiota, and infections have proven to have a significant role in epigenetic modifications. It is known that epigenetic modifications may cause behavioral and neuronal changes observed in neurodevelopmental disabilities, including fragile X syndrome (FXS) and autism (ASD). Probiotics are live microorganisms that provide health benefits when consumed, and in some cases are shown to decrease the chance of developing neurological disorders. Here, we examined the epigenetic outcomes in offspring mice after feeding of a probiotic organism, *Lactobacillus reuteri* (*L. reuteri*), to pregnant mother animals. In this study, we tested a cohort of Western diet-fed descendant mice exhibiting a high frequency of behavioral features and lower FMRP protein expression similar to what is observed in FXS in humans (described in a companion manuscript in this same GENES special topic issue). By investigating 17,735 CpG sites spanning the whole mouse genome, we characterized the epigenetic profile in two cohorts of mice descended from mothers treated and non-treated with *L. reuteri* to determine the effect of prenatal probiotic exposure on the prevention of FXS-like symptoms. We found several genes involved in different neurological pathways being differentially methylated (*p* ≤ 0.05) between the cohorts. Among the key functions, synaptogenesis, neurogenesis, synaptic modulation, synaptic transmission, reelin signaling pathway, promotion of specification and maturation of neurons, and long-term potentiation were observed. The results of this study are relevant as they could lead to a better understanding of the pathways involved in these disorders, to novel therapeutics approaches, and to the identification of potential biomarkers for early detection of these conditions.

## 1. Introduction

Several neurodevelopmental disabilities such as ASD cannot be well-characterized by the one-disease one-gene school. Their complex nature is attributed to the interplay between genetic, epigenetic alterations, neuroinflammation, and the innate immune system, that are associated with factors such as alteration in the gut microbiota and other environmental factors [1]. In the past decade or so, the interest of the research community started to increase towards epigenetic modifications such as DNA methylation, histone modifications, RNA interference, and their influence on gene expression [2]. Several studies have shown how different epigenetic modifications may cause behavioral and neuronal changes observed in neurodevelopmental disabilities [3,4,5,6]. Furthermore, it was shown that environmental factors such as diet, gut microbiota, and infections have significant roles in epigenetic modifications [7]. Specifically, studies have shown that bioactive nutrients and gut microbiota can alter the epigenetic profile via different mechanisms [8] and play a significant role in the regulation of the gut–brain axis [9]. The gut microbiota influences the epigenome by controlling the production of inflammatory cytokines in addition to antimicrobial peptides [10]. Short-chain fatty acids (SCFAs), synthesis of vitamins, neurotransmitters, and proper absorption of different nutrients are key functions of normal gut flora which help to maintain an intact and healthy gut–brain axis communication [11]. Importantly, a study showed that modulating the mother’s microbiome prenatally, or blocking a specific pro-inflammatory cytokine, interleukin 17-a, could prevent the development of autism-like neurodevelopmental disabilities in the mouse [12]. 

*Lactobacillus reuteri* (*L. reuteri*) is a well-studied probiotic bacterium that can colonize an array of mammals. In humans, it is found in different organs, including the gastrointestinal tract, urinary tract, skin, and in breast milk [13]. It has multiple benefits, such as strengthening the immune system and maintaining the balance of other beneficial microorganisms through the release of an antibacterial substance called “reuterin” [13]. 

Oral administration of *L. reuteri* in mice significantly reduced the pro-inflammatory cytokine response, specifically the percentage of Th17 cells and IL-17 levels [14]. 

Dysbiosis, which is the imbalanced gut microbiome with an increase in the population of gut bacteria with pathogenic traits, is frequently observed in children with neurodevelopmental disabilities [9,10]. Several studies have reported on the role of intestinal microorganisms in epigenetic alterations in terms of dysbiosis [9,10]. 

Fragile X syndrome (FXS), a model of epigenetic dysregulation, is the most common monogenic cause of ASD, with ~60% of FXS individuals presenting autism spectrum disorder [15,16]. FXS is caused by a long (>200) CGG trinucleotide repeat expansion, which suppresses the transcription via hypermethylation of the promoter and of the repeat located within the 5′-UTR of the fragile X messenger ribonucleoprotein 1 (*FMR1*) gene. The transcriptional silencing and the absence or reduced expression of the encoded gene product, FMRP [17], is what causes FXS. The epigenetic silencing of *FMR1*, specifically by DNA methylation and histone modifications resulting in loss of FMRP, has genome-wide consequences because of the role of FMRP in regulating the expression of several non-coding RNAs [18]. As a result, the transcription of many genes, playing a role in synaptic plasticity and neuronal functions, is affected in FXS.

Here, we report our findings from a study that characterized the global DNA methylation profile of biological samples derived from two groups of mice, those whose mothers had a prenatal intake of *L. reuteri* and those whose mothers did not. This study aimed to determine whether there is any potential interaction between the *L. reuteri* intake and changes in the methylomic profiles of one or both groups and if maternal induced diet correlated with the observed phenotype. The focus was on the pathways involved in neurogenesis and synaptogenesis, in addition to inflammation. 

The results of this study are relevant as they could lead to a better understanding of the pathways involved in neurodevelopmental disabilities, independently coming from some of the already identified risk genes, to novel therapeutics approaches, and to the identification of potential biomarkers for early detection of these conditions. 

## 2. Materials and Methods

### 2.1. Mouse Models

A cohort of 16 outbred CD1 Swiss mice, six months of age and of approximately the same weight, were used in this study. They were housed and handled in the Association for Assessment and Accreditation of Laboratory Animal Care-accredited facilities with diets, experimental methods, and housing as approved by the MIT Institutional Animal Care and Use Committee. Out of 16 mice, 8 were born from descendants of mothers which consumed a Westernized diet [19]; specifically, they were born to mothers who prenatally took *L. reuteri*, reported here as Group A, and 8 were born to mothers who did not prenatally take *L. reuteri*, and therefore received regular untreated water, reported as Group B. Mother’s drinking water was the only difference between the two groups. 

Half brains derived from the 16 mice were snap frozen and then pulverized and divided into aliquots. Genomic DNA (gDNA) samples were extracted from the brain tissue by standard procedure (Qiagen, Redwood, CA, USA) followed by quantification via a Qubit fluorometer (Invitrogen, Waltham, MA, USA). 

### 2.2. DNA Methylation Array Processing and Mapping MethylationEPIC Primer to the Mouse Genome

Five hundred ng of genomic DNA was bisulfite-treated using the EZ DNA methylation kit (Zymo research, Irvine, CA, USA). Although Infinium Mouse Methylation BeadChips have been developed recently to measure the methylation profile of mice, there was no mature mouse methylation array when this study started, and thus, the study was carried out using the Illumina Human Infinium Methylation BeadChips. Bisulfite-converted DNA was hybridized with the Illumina Human Infinium MethylationEPIC BeadChip Kit (Illumina, San Diego, CA, USA) and then scanned with the Illumina iScan System (Illumina, CA, USA) using the manufacturer’s standard protocol. Samples from the two groups were randomly loaded on the arrays. 

The Infinium MethylationEPIC BeadChip Kit included >850,000 individual CpG sites genome-wide at single-nucleotide resolution. As the Infinium MethylationEPIC array was not specifically designed for the mouse genome, many primers for each probe in the array do not target or specifically align within the mouse genome, so their location cannot be identified in the mouse genome with certainty. 

Hence, 22,569 out of the 850,000 probes that were identified in previous studies [20,21] to have a unique best alignment score when mapping the probe primers to the mouse genome (GRCm38) were used for the following analysis. 

### 2.3. Methylation Data Preprocessing

DNA methylation data were pre-processed and quantile-normalized with R package ‘minfi’ (Version 4.2.1, Bioconductor, Online Platform) [22]. Probes were filtered according to the following criteria: probes with a detection *p*-value greater than 0.01 in more than 5% of total samples (≥2). 

### 2.4. Identification of Differentially Methylated CpGs (DMs)

The Limma package was used to compare the methylation difference of CpGs that passed the quality control described above between the two groups. M value was used in statistical analysis instead of β value. M value is the ratio of the methylated to the unmethylated intensity and has a statistical advantage over the β value [23].

### 2.5. Pathway Analysis

Before the pathway analysis, the significance of each gene was determined. In the case of genes with one CpG site, the *p*-value of the CpG site was used as that of the gene. The *p*-values of genes with multiple CpG sites in the gene were calculated by combining the *p*-values of all CpG sites in the gene, using Fisher’s method [24]. 

R package ‘fgsea’ was used to carry out gene set enrichment analyses on gene sets based on Gene Ontology and metabolic pathways [25]. In Gene Ontology, gene sets are divided into three categories: biological process (BP), molecular function (MF), and cellular component (CC). The metabolic pathway was collected from four different sources: EHMN, HUMANCYC, KEGG, and REACTOME. 

### 2.6. Constructing a Functional Association Network for Some of the Proteins Involved in Brain Functions Using String 

Different proteins that are encoded by the genes involved in different brain functions based on the conducted pathway analysis were inserted into the String database to see the possible protein–protein interactions between them [26]. A set of proteins were entered in the String database, and an identifier mapping on the input was performed and displayed as a network covering all the mapped proteins and interconnections between them, if any. 

### 2.7. Western Blot Analysis

Lyophilized brain tissues from Group A of treated mice (*n* = 8) and Group B of untreated mice (*n* = 8) were homogenized with a tip-sonicator (Q-Sonica) in homogenization buffer containing 20 mM of HEPES buffer (Thermo Fisher Scientific, Waltham, MA, USA), 0.25 M of sucrose (MiliporeSigma, Burlington, MA, USA), and 1X protease inhibitor cocktail (MiliporeSigma, Burlington, MA, USA). Homogenates were centrifuged at 200,000× *g* rpm for 45 min at 4 °C and cytoplasmic fractions (supernatant) were collected and further quantified using the Bradford assay (BioRad Laboratories, Inc. Hercules, CA, USA). Ten µg of protein were loaded on 4–12% Bis-Tris gels (BioRad Laboratories, Inc. Hercules, CA, USA) and run at 100 V for 60 min and then 130 V for 60 min. Resolved proteins were then transferred onto nitrocellulose membranes using the Trans-Blot Turbo transfer system (BioRad Laboratories, Inc. Hercules, CA, USA) at 25 V, 1.0 A, for 30 min. Membranes were stained with Ponceau to test for transfer efficiency, blocked with 3% milk for 1 h at room temperature, followed by incubation with 1:500 diluted Agap3 primary antibodies (sc-390362, Santa Cruz Biotechnology, Inc. Dallas, TX, USA), 1:2000 diluted Shank3 primary antibodies (ab 264347, abcam, Cambridge, MA, USA), and 2.7:1000 diluted Dlg2 primary antibodies (ab 261634, abcam, Cambridge, MA, USA), respectively, overnight at 4 °C. Membranes were then washed in 1X-TBS and incubated with HRP-linked secondary antibody diluted 1:10,000 (Catalog # 1706516, BioRad Laboratories, Inc., Hercules, CA, USA) for Agap3, and HRP-linked secondary antibody diluted 1:2000 (cell signaling, Catalog 7074) for Shank3 and Dlg2, for 1 h at room temperature. Bands were then visualized using a Chemiluminescent substrate, Super Signal West Dura (Thermo Fisher Scientific, Waltham, MA, USA). Densitometry analysis of bands for relative protein quantification was performed using the α Innotech Gel Imaging System (Cambridge Scientific, Watertown, MA, USA).

## 3. Results

### 3.1. Analysis

#### 3.1.1. Mouse Models

Group A mice were male progeny that received probiotic *L. reuteri* 6475 exposures in utero, and they were clinically normal. Group B mice were male progeny whose mothers received placebo water prenatally. Group B mice had FXS-like symptoms including physical features such as dimorphic head, ears, and enlarged testicles, and behavioral features such as head bobbing, hyperactivity, and stereotypy, similarly to the typical features observed in humans with FXS. 

#### 3.1.2. Identification of Perfectly Mapping Probes to the Mouse Genome

Although 128,259 probes passed QC, we only included 17,735 of them with primers that had high mapping quality to the mouse genome for the following analysis (Appendix A). The β value distribution of all probes that passed QC has a large peak around 0.25 (Appendix A), while the β value distribution of probes that passed quality control and had a unique best alignment score has two peaks, one around 0 and the other around 0.9 (Appendix A), similar to what was reported by Garcia-Prieto et al. [27]. The probes that passed QC without a unique best alignment score were excluded from the analysis, as they were likely targeting multiple locations, making the measurement of the levels of methylation not reliable.

#### 3.1.3. Identification and Distribution of DMs among the Two Groups

Between the 2 groups, 570 probes were found to be differentially methylated (*p*-value < 0.05, Appendix A). The distribution of the top 100 CpGs with the largest variations is shown in Figure 1. Interestingly, four samples (20-473, 20-675, 20-676, and 20-678) in Group B showed a very different CpG methylation profile than the other samples in Group B and those in Group A.

#### 3.1.4. Differentially Methylated Genes in FXS-Like Mice Are Involved in Brain Functions 

DMs were mapped to several genes involved in different neurological pathways (*p* ≤ 0.05). Among the key functions, synaptogenesis, neurogenesis, synaptic modulation, synaptic transmission, reelin signaling pathway, promotion of specification and maturation of neurons, and long-term potentiation were observed. Furthermore, additional genes implicated in schizophrenia, intellectual disabilities, social deficits, repetitive behaviors, deafness, pituitary adenomas, and dysmorphic facies were also identified (Table 1).

#### 3.1.5. Genes and Pathway Enrichment Analysis

Methylation analysis was obtained from 17,735 sites for the whole mouse genome, which identified DMs located in 6418 genes. The results of the Gene Ontology enrichment analysis datasets shown in Figure 2 indicate that several pathways were significantly enriched between the two groups of mice. As for the biological processes (BP), among the top pathways was that involved in the activation of cysteine-type endopeptidase activity involved in apoptotic processes (GO:0006919). Another pathway was that involved in retrograde transport of membrane-bounded vesicles from endosomes back to the trans-Golgi network, where they are recycled for further rounds of transport (GO:0042147). The cellular component that was found to be of significance was the endocytic vesicle, which is formed by invagination of the plasma membrane around an extracellular substance and fuses with early endosomes to deliver the cargo for further sorting (GO:0030139)). The last subset of Gene Ontology studies was related to molecular function (MF), with the most significant being the ligation activity of acids to amino acids with the concomitant hydrolysis of ATP (GO:0016881). 

In addition, there was an array of statistically significant pathways, including a positive regulation of MAPK activity (*p* = 0.005), embryonic placenta development (*p* = 0.009), and the positive regulation of inflammatory response (*p* = 0.015). A list of GO pathways that were found to be of significance and the corresponding genes involved are listed in Appendix A. We focused mainly on pathways that are involved in brain development and maintenance of the central nervous system. A total of five pathways were found to be statistically significant, including midbrain development (*p* = 0.004), positive regulation of neuron apoptotic process (*p* = 0.01), negative regulation of neuron projection development (*p*-value = 0.02), cerebral cortex neuron differentiation (*p* = 0.04), and neuromuscular junction development (*p* = 0.05) (Table 2).

#### 3.1.6. String-Based Protein–Protein Interaction Network Analysis

Several proteins that were identified through the pathway enrichment analysis to be involved in different brain functions were found to interact together. Some of those proteins were found to interact with several others, such as LIM homeobox transcription factor 1-β (LMX1B) which interacts with Homeobox protein OTX2 (OTX2), Sonic hedgehog protein (SHH), Homeobox protein MSX-1 (MSX1), and Homeobox protein engrailed-1 (EN1). Another protein that was found to interact with other different proteins involved in different functions in the brain is Fibroblast growth factor receptor 2 (FGFR2), which interacts with MSX1, SHH, Wingless-type MMTV integration site family, member 1 (WNT1), and Fibroblast growth factor receptor 1 (FGFR1) (Figure 3).

#### 3.1.7. DLG2, SHANK3, and AGAP3 Protein Expression

To validate our methylation findings, among the differentially methylated genes between the two groups, we measured the expression level of a subset of the correspondent encoded proteins, including AGAP3, SHANK3, and DLG2. Western blot analysis in the group of mice treated with probiotics (Group A, *n* = 8) and in those without any treatment (Group B, *n* = 8) revealed a significant difference in expression levels of DLG2 between both groups (*p* = 0.026), while the difference in AGAP3 and SHANK3 expression levels did not reach significance (Figure 4). 

## 4. Discussion

During pregnancy and in early childhood, many factors such as hormones, stress, genetics, and diet can affect the brain development of a child. Studies in recent years have demonstrated that gut microbiota and maternal obesity can also influence a child’s neurodevelopment [54]. Changes in the microbiota profile that occur within the maternal gut could affect the metabolic profile of the mother and contribute to the immune system and metabolism of their children [55]. Furthermore, microbiota establishment and neurodevelopment were found to be vulnerable to damage during similar developmental windows, with altered developing gut microbiota therefore affecting neurodevelopment and potentially leading to mental health problems [56]. Studies have also shown that mothers who have experienced maternal immune activation (MIA) have an increased risk of their offspring having ASD [57]. 

Animal experiments indicate that interleukin-6 cytokines (IL-6) are key to the induction of ASD by MIA [58,59,60] and that increased levels of both IL-6 and IL-17a in maternal serum and fetal brains were halted by oral probiotics, as were the loss of parvalbumin-positive neurons and the declines in γ-aminobutyric acid levels in adult offspring brain [61]. The metagenomic whole shotgun sequencing of maternal high-fat diet (MHFD) fecal microbiota showed a significant reduction in the abundance of the commensal *L. reuteri* [62,63], while synaptic plasticity within the ventral tegmental area, hypothalamic oxytocin, and social behavior in maternal high-fat diet offspring were restored by microbial reconstitution with *L. reuteri* [64].

In this study, we compared the epigenetic profile of two groups of mice: those that were exposed to *L. reuteri* in utero and those that were exposed to placebo. A total of 23 genes known to have a role in neurological function (Table 1) showed a significant differential methylation between the two groups. We recognize that we cannot rule out the possibility that observed methylation differences may stem from the genetic variability introduced by our mouse model [65]. However, given the significant differences found across control and experimental groups testing randomly sorted litter mates, our results are likely to be ascribable to an altered gut microbiota. 

Among them were the *Ncdn*, the *Agpat1*, the *Dctn1*, the *Fbx011*, the *Emx2*, the *Dcc* gene, and the *Gart* genes, all encoding for proteins associated with seizures, abnormal hippocampal brain development ID, ASD, ADHD, anxiety, developmental delays, aggression, hyperactivity, altered social skills, delays in speech and language skills, and childhood epilepsy [46,51,66,67]. A number of identified genes were associated with ASD and listed in the SFARI database (Table 1, labeled with an asterisk). A study showed that a mouse model with the Pax6 mutation presented with aggression in the reciprocal social interaction test and hyperactivity signs in the open-field test and the tail suspension test [68]. The *Dlgap2* KO mice had deficits in learning, abnormal social behavior, and intense aggressive behavior, which are behavioral features observed in both ASD and FXS [63,64,69]. 

Among the identified differentially methylated genes, the *Dlg2*, *Shank3,* and *Agap3* were significantly hypermethylated in the FXS-like group compared to the control group (Table 1). 

*Dlg2* deficiency in mice contributes to aberrant synaptic transmission accompanied by reduced sociability and increased repetitive behavior [35]. In neurons, specific *Dlg2* isoforms bind to NMDA (N-methyl-aspartate) receptors and K channels, mediating their clustering on the postsynaptic membrane, and they are differentially expressed in plasmacytoid dendritic cells (pDCs), potentially having different functions in the neuronal postsynaptic membrane [36]. Furthermore, two very similar proteins, PSD-93 and PSD-95, which developed from a duplication of a single ancestral gene, play an opposite role in the maturation of synapses from immature to active, with PSD-95 promoting maturation, while PSD-93 slows it down [34]. Thus, fine-tuning and a balance between these proteins appear to be essential during developmental critical periods, suggesting that impairment in their expression in either direction could be relevant or constitutes a potential leading cause of neurodevelopmental disorders. 

SHANK3, highly expressed in both the peripheral and the central nervous system, is a scaffolding protein, which plays a key role in synaptic development and function. Impaired gene expression, mutations, methylation, and alternative splicing of SHANK3 have been reported in ASD [41]. Moreover, SHANK3-haploinsufficiency results in altered function, which negatively affects synaptic development and function, and its disruption contributes to ASD etiology and is also observed in individuals affected by Phelan-McDermid syndrome [70,71,72]. In this study, we observed hypermethylation, lower protein expression levels, and thus haploinsufficiency of SHANK3 expression in the FXS-like group compared to the controls, which strongly suggest its potential role in the phenotype displayed by these mice. 

Centaurin-gamma3, or AGAP3, is an essential signaling component of the NMDA receptor complex that links NMDA receptor activation to AMPA receptor trafficking [28]. Furthermore, a study conducted by Gross et al. [25] showed that PIKE reduction rescued PI3K-dependent and -independent neuronal defects in FXS. Furthermore, it was determined that the genetic reduction of *CenG1A*, Drosophila’s ortholog of PIKE, rescued excessive PI3K signaling and mushroom body defects [29]. In this study, even though *CenG1A* was hypermethylated, its protein levels were higher in the FXS-like group compared to the control group. This might be another factor contributing to the development of FXS-like symptoms in that group, especially since it was shown in the above study that the increased expression of PIKE, which is encoded by *CenG1A*, mediates deficits in synaptic plasticity and behavior in Fragile X syndrome [29].

On the other hand, some genes were not associated with ASD nor FXS but had significant roles in brain functions. For example, *Nbea* is implicated in developmental diseases with early generalized epilepsy phenotypes [49]. Overexpression of *Gart* improves the IQ of patients with Down syndrome [50]. Hypermethylation of *Agap1* is associated with neurodevelopmental disorders through interference with the cellular structure and internal functioning of neurons and through interactions with other genes, particularly the *Dtnbp1* gene [52]. Considering the neurological disorders that are connected to hypermethylation of *Agap1*, one potential impact of *L. reuteri* that can be inferred from the presented mouse experiment is that it reduced the methylation of *Agap1* in mice that received the probiotic in utero, making the mice more neurotypical. In addition, hypermethylation of *Agpat1,* as well as its loss-of-function mutations, have been found to correlate with metabolic and reproductive disorders, seizures, and abnormal hippocampal brain development [66]. *Fbxo11* was shown to be involved in several neurological disorders. Damage to this gene correlates with intellectual disabilities, autism spectrum disorder, ADHD, and anxiety [67].

Thus, these findings may help to explain why these different proteins interact together and might collectively contribute to the observed FXS-like symptoms in the cohort of mice included in this study. 

It is believed that administration of *L. reuteri* might be transferred from mothers to pups, which could halt alteration in the expression of the genes discussed, hence preventing the development of FXS-like symptoms in the offspring. In our studies (see the companion manuscript in this same issue), C-section rederivation with cross-fostering was used to eliminate the possibility of mother-to-infant transfer of microbiota. 

Our findings are of importance as the observed Fragile X-like syndrome (FXLS), in mice, is based on the maternal microbiome, rather than the presence of the *Fmr1* CGG expansion, and because a link between maternal gut dysbiosis and the development of FXS has never been reported. 

Thus, the gut microbiome might contribute to the development of behavioral and physical features associated with FXS, although the mechanisms by which altered intestinal bacteria may affect brain development and function are not currently known.

## Figures and Tables

**Figure 1 genes-13-01300-f001:**
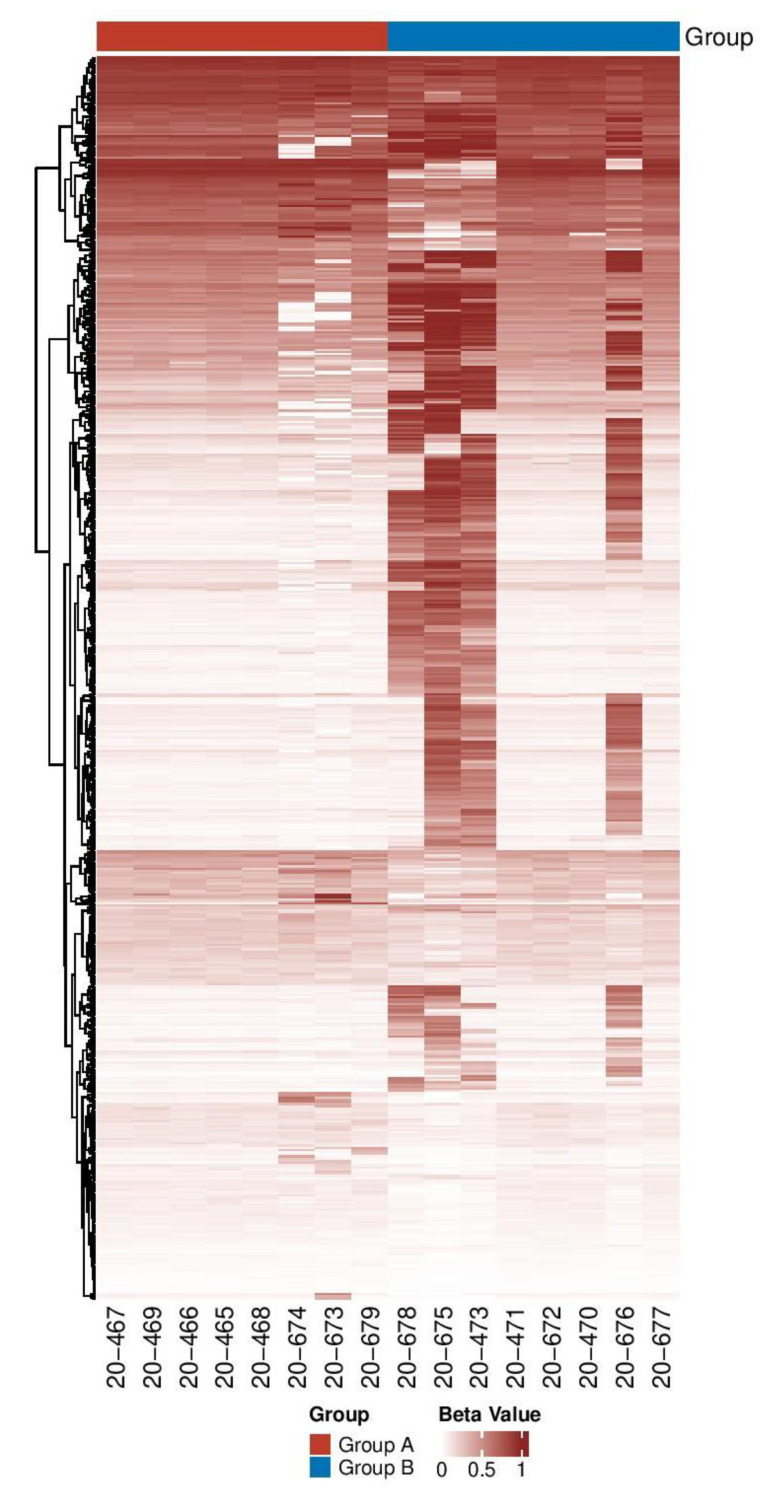
Heat map showing both Groups A (TD) and B (FXS-like symptoms) methylation patterns. As the color gets darker, this means more hypermethylation. The heatmap is showing the β values from 570 significant CpGs. CpGs were hierarchically clustered.

**Figure 2 genes-13-01300-f002:**
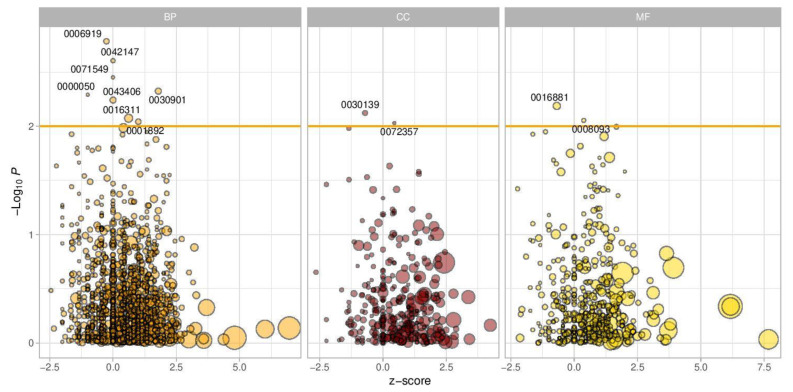
Gene Ontology analysis. Volcano plot of results from enrichment analysis of pathways from Gene Ontology. Z-score is defined as the number of hypermethylated genes minus the number of hypomethylated genes in the pathway, divided by the square root of the number of total genes. A positive Z-score indicates hypermethylation in Group B, while a negative Z-score indicates hypomethylation in Group B. Orange vertical line indicates a *p*-value of 0.01. Gene sets with a *p*-value less than 0.01 are labeled with GO ids. The point size shows the number of genes with methylation information in each pathway. BP: biological process: CC: cellular component; MF: molecular function.

**Figure 3 genes-13-01300-f003:**
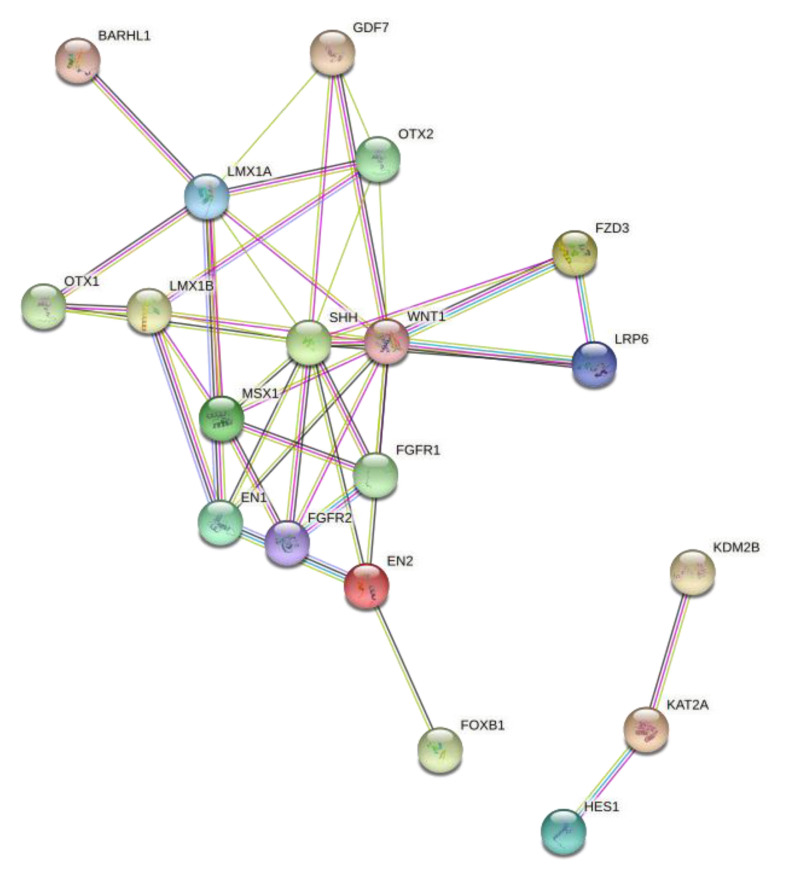
Protein–protein interactions network. Protein interactome network for the proteins that were shown to be differentially methylated in FXS-like mouse models compared to controls using the STRING software. All the colored nodes represent the query proteins and first shell of interactions. Each node represents all the proteins produced by a single, protein-coding gene locus. Filled nodes mean that the 3D structure of that protein is known or predicted. The colored lines represent different types of associations. The navy-blue line indicates known interactions from curated databases. The fuchsia lines indicate known interactions that are experimentally determined. The dark green lines represent predicted interactions of neighboring genes, the red lines represent gene fusions, and the royal blue line represents gene co-occurrence. The light green line represents text mining, the black line represents co-expression, and the purple line represents protein homology.

**Figure 4 genes-13-01300-f004:**
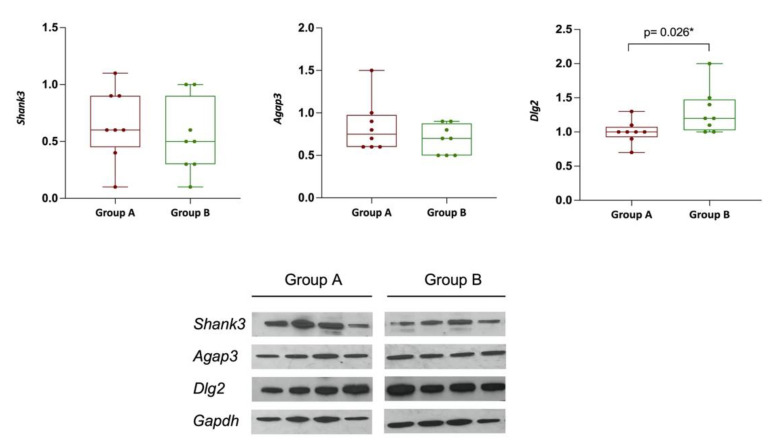
(**Top**): Boxplot charts showing the differential expression levels of SHANK3, AGAP3, and DLG2 in the two groups of mice. The comparison of expression levels of both SHANK3 and AGAP3, although in the right direction relative to the methylation data, did not reach statistical significance. A significant increased expression of DLG2 was observed in Group B compared to Group A. (**Bottom**): Western blot of representative brain samples from Groups A and B (*n* = 4) for all three proteins. GADPH was used as a loading control. * Denotes statistical significance (*p*-value < 0.05).

**Table 1 genes-13-01300-t001:** List of differentially methylated genes involved in several neurological functions. The genes labeled with * are the ones associated with ASD and listed in the SFARI database.

Gene	*p*-Value	Me Status	Function/Implication
*Agap3 **	0.008	Hypermethylated	Regulates NMDA receptor-mediated Ras/ERK and Arf6 signaling pathways in *LTP* [28,29]
*Pax6 **	0.004	Hypermethylated	Regulates cortical progenitor proliferation, neurogenesis, and formation of cortical layers [30]
*Zbtb20 **	0.01	Hypermethylated	Plays a role in dendritic and synaptic structure [31]
*Tfap2a*	0.007	Hypermethylated	Promotes specification and maturation of neurons [32]
*Kcnn2*	0.02	Hypermethylated	Plays a role in synaptic transmission [33]
*Dlg2 **	0.01	Hypermethylated	A DLG2 deficiency in mice leads to reduced sociability and increased repetitive behavior accompanied by aberrant synaptic transmission [34,35,36]
*Clasp2 **	0.03	Hypermethylated	Plays an important role in the reelin signaling pathway [37]
*Dlgap3 **	0.03	Hypomethylated	Important for protein–protein interactions at synapses and transmission across chemical synapses. Implicated in body dysmorphic disorders [38]
*CNot3 **	0.04	Hypomethylated	Diseases associated with CNOT3 include intellectual developmental disorder with speech delay, autism, and dysmorphic facies [39]
*Cdh23*	0.05	Hypomethylated	Implicated in pituitary adenoma, deafness [40]
*Shank3 **	0.01	Hypermethylated	Implicated in schizophrenia, ASD, and other neurological disorders [41].
*Abl1*	0.03	Hypermethylated	Important for assembly and remodeling of synapses [42]
*Ap2a1*	0.01	Hypomethylated	AP-2 seems to play a role in the recycling of synaptic vesicle membranes [43]
*APC*	0.02	Hypomethylated	Regulates synaptic adhesion and signal transduction pathways critical for normal cognition and behavior [44]
*Kmt2d **	0.03	Hypermethylated	Promotes transcriptional activation and its loss causes the intellectual disability disorder Kabuki syndrome 1 (KS1) [45]
*Ncdn*	0.02	Hypermethylated	Neurodevelopmental delay, intellectual disability, and epilepsy [46]
*Dctna*	0.03	Hypermethylated	Stabilizing neuron cytoskeleton [47]
*Gnas **	0.04	Hypermethylated	Implicated in variable degrees of intellectual disability and developmental delay [48].
*Nbea **	0.04	Hypermethylated	Developmental disease gene with early generalized epilepsy phenotypes [49].
*Gart*	0.04	Hypermethylated	Overexpression improves the IQ [50]
*Emx2*	0.05	Hypermethylated	Schizencephaly, CNS, tumorigenesis [51]
*Agap1 **	0.03	Hypermethylated	Dendritic spines, neuron endosome trafficking, neurodevelopmental disorders [52]
*Dcc **	0.04	Hypermethylated	Mirror movement, gaze palsy, impaired intellectual disability [53]

**Table 2 genes-13-01300-t002:** GO pathways, and the corresponding genes involved in these pathways.

Description	Number of Genes	Z-Score	*p*-Value	Genes
GO_BP_MM_MIDBRAIN_DEVELOPMENT	22	1.788854382	0.004	*Barhl1*;*Msx1*;*Kdm2b*;*Lrp6*;*Foxb1*;*Otx1*;*Otx2*;*Fgfr2*;*En2*;*Lmx1b*;*En1*;*Aplp2*;*Lmx1a*;*Rfx4*;*Gdf7*;*Hes1*;*Fgfr1*;*Shh*;*Fzd3*;*Tal2*;*Kat2a*;*Wnt1*
GO_BP_MM_POSITIVE_REGULATION_OF_NEURON_APOPTOTIC_PROCESS	19	1.697749375	0.01	*Tfap2a*;*Cdk5*;*Map3k11*;*Nf1*;*Tfap2b*;*Bax*;*Hrk*;*Jun*;*Cdk5r1*;*Ptprf*;*Map2k7*;*Trp53*;*Ascl1*;*Agrn*;*Srpk2*;*Tgfb2*;*Nr3c1*;*Epha7*;*Ube2m*
GO_BP_MM_NEGATIVE_REGULATION_OF_NEURON_PROJECTION_DEVELOPMENT	16	1.069044968	0.02	*Pafah1b1*; *Cbfa2t2*;*Bcl11a*;*Gfap*;*Ntm*;*Dpysl3*;*Runx1t1*;*Vim*;*Inppl1*;*Gfi1*;*Fkbp4*;*Trpv4*;*Itm2c*;*Inpp5j*;*Lpar1*;*Rtn4*
GO_BP_MM_CEREBRAL_CORTEX_NEURON_DIFFERENTIATION	6	0.447213595	0.04	*Pafah1b1*;*Nkx2-1*;*Id4*;*Lhx6*;*Nr2e1*;*Pex5*
GO_BP_MM_NEUROMUSCULAR_JUNCTION_DEVELOPMENT	14	–0.632455532	0.05	*Cacnb4*;*Fgfr2*;*Cacna1s*;*Col4a1*;*Cacna2d2*;*Pdzrn3*;*Cacng2*;*Lamb2*;*Agrn*;*Pak1*; *Dvl1*;*Erbb2*;*Tnc*;*Ky*

## Data Availability

Data and results generated from this project will be fully available upon request. Biological samples from subjects included in this study will be available under MTA agreement accordingly to the University of California, Davis policy.

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
