# Peer review of "Differential Methylation Profile in Fragile X Syndrome-Prone Offspring Mice after in Utero Exposure to Lactobacillus Reuteri"

_genes, 2022, doi:10.3390/genes13081300_

Round 1
Reviewer 1 Report
In this manuscript AlOlaby et al. describe methylation profile and differences among two groups of mice after in utero exposure to a probiotic diet. The data presented is of interest; however there is one aspect that needs to be addressed in order to understand the comparison performed. Which is the genotype of the mice used? Are the mothers of both groups identical genetically? What about the mice descendant? Are they KO for the FMR1 gene? It has been described that episignatures are associated with genetic syndromes, thus the two groups of mice should be identical in order to assure that methylation differences are due to probiotic diet.
A part from this major concern there are some minor aspects that need to be addressed:
Check for references citation
Check how genes names for mice are written (Gene symbols should be italicized, with only the first letter in upper-case)
Author Response
Thank you so much for your valuable input. We truly appreciate your comments.
Kindly find the responses below:
Which is the genotype of the mice used?
This wild-type mouse model is described in greater detail in a companion manuscript Maternal microbiota modulate a Fragile X-like syndrome in offspring mice also submitted to this GENES special topic issue. Briefly, the mice are all wild-type outbred CD-1 stock in a large multigenerational study. In this case, predispositions to a Fragile X Syndrome-like phenotype were not a consequence of genetic engineering, rather to gut microbiome alterations.
Are the mothers of both groups identical genetically? What about the mice descendant?
As described above, these mice are part of a large multigenerational study using randomly subdivided wild-type outbred CD-1 stock littermate siblings for each test group. They are not genetically identical.
A powerful aspect of this design is that all animals are descendants of mice having fast-food style diets in earlier generations, intending to mimic human Westernized fast-food consumption and consequent microbiome alterations described in earlier publications. Like humans these mice were not genetically identical, but they nonetheless had significantly different phenotypes and epigenetic signatures after the probiotic.
Are they KO for the FMR1 gene?
No, the FMR1 gene is intact.
It has been described that episignatures are associated with genetic syndromes, thus the two groups of mice should be identical to assure that methylation differences are due to probiotic diet.
We recognize that we cannot entirely rule out the possibility that observed methylation differences could stem from the genetic variability introduced by our model, and have added this statement under our Discussion. However, given the consistent, significant differences found across our control and experimental groups, we believe our results are ascribable to the altered microbiome.
While methylation pattern differences are conventionally studied in genetically identical inbred strains, our use of outbred stock mice allows us to simulate disease conditions more closely resembling human genetic diversity. Thus, statistically significant outcomes in the face of genetic diversity may offer enhanced experimental replicability1 and yield more translatable results. Outbred mice also produce larger numbers of viable offspring, which is particularly advantageous for multigenerational experiments such as those in these studies.
- Tuttle et al. 2018. Comparing phenotypic variation between inbred and outbred mice. Nature Methods 15, 994–996. doi https://doi.org/10.1038/s41592-018-0224-7
Apart from this major concern there are some minor aspects that need to be addressed:
Check for references citation
References have been checked
Check how genes names for mice are written (Gene symbols should be italicized, with only the first letter in upper-case)
Gene names have been checked

Reviewer 2 Report
Dear Authors,
your paper discuss about the role of maternal gut microbiota during pregnancy and brain development and function in offspring mice. Although, your MS presents very interesting hints, it has some hasty conclusions.
Major comments:
1. the fact that some mice in the group B present with stereotypies, this does not necessarily unite them to FXS, because stereotypies are findings common to different forms of neurodevelopmental disorders and ASD (section 3.1.1). Epigenetic modifications are not the only event occurring in FXS, maybe they are consequence of CGG expansion (see Abstract lines 16-18). Furthermore, mouse is not the best model to study epigenetics in FXS; it would have been different to study the effect of gut microbiota in Fmr1KO mice. Lower FMRP expression (Abstract line 23) should be demonstrated through a WB (or other technique) in a subgroup of mice from Group B.
Given all these considerations, conclusions (and also great emphasis in the title) about prevention of FXS-like phenotype in offspring after L. reuteri administration during pregnancy are unsupported by data presented in your paper. It is risky to speak of FXS-like group to refer to effectively normal mice (even if talking about mice!).
2. Usually DNA methylation levels (in a specific locus) are compared with the levels of gene expression and not with protein expression. A comment about this choice could be added. Again, from section 3.1.7 SHANK3 and AGAP3 protein levels seem similar in both groups of mice. In Discussion lower SHANK3 protein levels are reported in FXS-like subgroup (lines 371-373). This is a conflicting statement. Finally, a WB should be displayed to support data of section 3.1.7. The sentence "significant difference in expression levels of DLG2" could be better explained (higher or lower in a goup compared to the other).
3. In the Discussion could be included a statement about the choice and the adaptation of a human methylation platform to study methylation in mouse. The Discussion is too long and lists too many proteins, it could be shortened.
Minor comments:
- Figure 1 is not cited in the text, while Figure 4 is missing.
- References should be verified both in the text and in the list (sometimes they are numbered, while other are cited as "Author Name and year of publication").
- Grammatical errors and typos should be checked after a careful re-reading.
Author Response
Thank you so much for your valuable comments. We truly appreciate it.
Kindly find below the responses:
REVIEWER 2: Dear Authors, your paper discuss about the role of maternal gut microbiota during pregnancy and brain development and function in offspring mice. Although, your MS presents very interesting hints, it has some hasty conclusions.
Major comments:
The fact that some mice in the group B present with stereotypies, this does not necessarily unite them to FXS, because stereotypies are findings common to different forms of neurodevelopmental disorders and ASD (section 3.1.1). Epigenetic modifications are not the only event occurring in FXS, maybe they are consequence of CGG expansion (see Abstract lines 16-18). Furthermore, mouse is not the best model to study epigenetics in FXS; it would have been different to study the effect of gut microbiota in Fmr1KO mice. Lower FMRP expression (Abstract line 23) should be demonstrated through a WB (or other technique) in a subgroup of mice from Group B.
We agree with the reviewer that there is a commonality and phenotype convergence across different neurodevelopmental disorders and that epigenetic modifications are not the only event occurring in FXS; indeed, point mutations and deletions leading to FXS have been reported. Studying the effect of gut microbiota in Fmr1KO mice is definitely interesting, and it will be carried out in future studies, but it was not the focus of this study. We have demonstrated lower FMRP expression by Western Blot analysis (please find attached a representative example of WB for the reviewer - Western blot were run in triplicates – three independent experiments). This data was not included in this manuscript as it is part of the companion manuscript submitted to this same special issue.
Given all these considerations, conclusions (and great emphasis in the title) about prevention of FXS-like phenotype in offspring after L. reuteri administration during pregnancy are unsupported by data presented in your paper. It is risky to speak of FXS like group to refer to effectively normal mice (even if talking about mice!).
We are not stating that these mice are like Fragile X mice, as they do not have a mutation in the FMR1 gene, but they present with an FXS-like phenotype (FXS-like symptoms). Similarly, we and others have previously reported on a Prader Willi-like phenotype in Fragile X, in absence of the 15qdel.
Usually DNA methylation levels (in a specific locus) are compared with the levels of gene expression and not with protein expression. A comment about this choice could be added.
Although it would have been nice to check if differential methylation between the two groups corresponded to differential transcriptional levels of the identified genes, due to availability of samples, we choose to check protein levels, which altered expression is what ultimately suggests their potential role in the observed phenotype.
Again, from section 3.1.7 SHANK3 and AGAP3 protein levels seem similar in both groups of mice. In Discussion lower SHANK3 protein levels are reported in FXS-like subgroup (lines 371-373). This is a conflicting statement. Finally, a WB should be displayed to support data of section 3.1.7. The sentence "significant difference in expression levels of DLG2" could be better explained (higher or lower in a group compared to the other).
We have added Figure 4, which include the Western blot data for the three proteins and clarified the difference in expression levels in the Results and in the Discussion sections.
- In the Discussion could be included a statement about the choice and the adaptation of a human methylation platform to study methylation in mouse. The Discussion is too long and lists too many proteins, it could be shortened.
Discussion has been modified and shortened. A comment about the human platform is included in the method section 2.2.
Minor comments:
- Figure 1 is not cited in the text, while Figure 4 is missing.
Figures are now correctly cited and included in the manuscript.
- References should be verified both in the text and in the list (sometimes they are numbered, while other are cited as "Author Name and year of publication").
References have been verified for format
- Grammatical errors and typos should be checked after a careful re-reading.
Spelling and grammar checking has been done

Round 2
Reviewer 1 Report
The authors have addressed all reviewer comments.
Reviewer 2 Report
Dear Authors,
thank you for your reply.
Your paper may be considered acceptable for publication in the present form.